# Granular Information Bottleneck for Deep Multi-modal Clustering

## Abstract

Deep multi-modal clustering generally focuses on improving clustering accuracy by leveraging information from different modalities. However, existing methods are designed around the finest-grained points as input, which are neither efficient nor robust to noisy data, negatively affecting the clustering results. To this end, we propose a novel granular information bottleneck (GIB) for deep multi-modal clustering, which embeds a dual-tiered information bottleneck constraint mechanism operating synergistically at the granular and sample levels, thereby learning discriminative feature representations with enhanced inter-cluster separability. Specifically, GIB adaptively represents and covers the sample points through granular balls of different granularity levels, which effectively captures the feature distribution within each cluster. Simultaneously, information compression and preservation are used to exploit the independence and complementarity of modalities while optimizing cluster assignments alignment. Finally, the objectives of GIB are formulated as a target function based on mutual information, and we propose a variational optimization method to ensure its convergence. Extensive experimental results validate the effectiveness of the proposed GIB model in accuracy, reliability and robustness.

## 1 Introduction

Multi-modal clustering has emerged as a powerful technique for integrating data from images, text, audio and other modalities to overcome the limitations inherent in single-modality methods and achieve superior clustering performance (Xia et al., 2022; Yang et al., 2022; Hu et al., 2024b). It has been widely applied in various real-world scenarios, including intelligent recommendation Liu et al. (2021), cross-modal retrieval (Chun et al., 2021; Hu et al., 2019; Yuan et al., 2022), and biomedical science (Si et al., 2023; Acosta et al., 2022).

**Deep Multi-modal Clustering.** Recent advances in deep multi-modal clustering (DMC) have significantly enhanced the representation quality and clustering performance through deeply exploring the complex interrelationships between modalities through powerful hierarchical nonlinear mapping (Caron et al., 2018; Palumbo et al., 2024; Liu et al., 2024; Jia et al., 2025a; Chu et al., 2024). Existing DMC methods can be roughly divided into three categories: *(1) Modality representation learning* (Li & Liao, 2021; Peng & Qi, 2019; Yang et al., 2017) which focuses on unified and modality-specific representations using techniques like adversarial embedding and adaptive fusion weights. For instance, Li & Liao (2021) disentangles and integrates information via a learned fusion strategy. *(2) Contrastive learning* (Hu et al., 2023; Lou et al., 2025; Zou et al., 2024; 2025) which improves representation discrimination by learning sample similarities and differences. Specifically, Zou et al. (2025) employs dual global information guidance for noise reduction and partition refinement. *(3) Graph-based multi-modal clustering* (Tan et al., 2023; Huang et al., 2022; Pan & Kang, 2021) which constructs graphs to model relationships between data points and uses graph neural networks for clustering. For example, Huang et al. (2022) improves accuracy by identifying consistency and discrepancies across graphs to form a structured consensus graph.

However, the above DMC methods suffer from escalating limitations that impair their handling of complex data. First, they process individual samples as input, rendering them vulnerable to noise, which distorts true distributions and impedes discriminative feature extraction Sun et al. (2024). As data scale and complexity grow, this exacerbates computational costs and challenges in model optimization, feature extraction, cross-modal alignment, and robustness enhancement. Consequently,

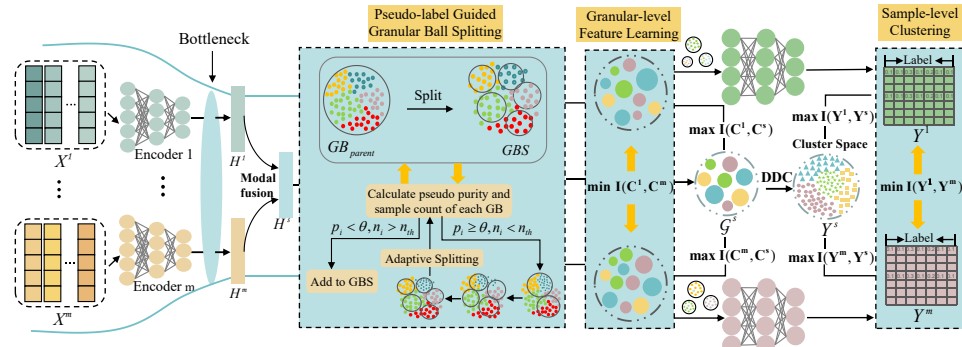

Figure 1: Illustration of the GIB framework. First, the GIB extracts modality-specific latent representations $\{H^i\}_{i=1}^m$ from multi-modal inputs $\{X^i\}_{i=1}^m$ using $m$ shared encoders, and then fuses them into a global representation $H^s$. Second, granular ball structures $\{\mathcal{G}^i\}_{i=1}^m$ and $\mathcal{G}^s$ are then processed by the DDC module to obtain modality-specific assignments $\{Y^i\}_{i=1}^m$ and a global assignment $Y^s$. Here, $\{C^i\}_{i=1}^m$ denote granular-ball centers for each modality, and $C^s$ is the granular ball centers from the fused features. Then, at the sample clustering alignment level, we aims to minimize $\sum_{i=1}^m I(X^i, H^i)$ and $\sum_{i=1}^m \sum_{j=i+1}^m I(Y^i, Y^j)$ to reduce redundancy and inconsistency, while maximizing $\sum_{i=1}^m I(Y^i, Y^s)$ for clustering alignment. Finally, at the granular feature learning level minimizes $\sum_{i=1}^m \sum_{j=i+1}^m I(C^i, C^j)$ to promote complementary representations and maximizes $\sum_{i=1}^m I(C^i, C^s)$ for modality-specific features alignment to the global representation.

these issues foster local optima and unstable training, diminishing overall efficiency. Finally, many existing methods overlook the inherent structure and semantic information of data, undermining cluster interpretability.

**Granular-ball-based Clustering.** Xia et al. (2023) proposed granular-ball computing as an innovative modeling approach within the field of multi-granularity cognitive computing. It is rooted in human cognitive mechanisms characterized by large-scale priority features Chen (1982). Building on the principles of granular computing (Xia et al., 2023; 2025; Huang et al., 2025), researchers apply it to data clustering. For example, Jia et al. (2025b) introduces granular-ball generation to enhance clustering quality and accuracy. Similarly, Xie et al. (2025) employs feature-weighted granular-ball graphs with Graph Convolutional Network (GCN) autoencoders for graph clustering. Moreover, Xie et al. (2023) proposes granular-ball spectral clustering to reduce computational time and resources. Additionally, Su et al. (2025) employs multi-view granular-ball contrastive clustering to address false negatives and capture local structures.

Although those methods have achieved remarkable clustering performance, they have two drawbacks. First, most current granular-ball clustering methods target single-modal data and struggle to capture diverse latent structures in multi-modal datasets, leading to inaccurate results in complex scenarios. More critically, the only existing multi-modal granular-ball method Su et al. (2025) overlooks inter-modal complementary relationships and clustering assignment guidance, while unconstrained sample numbers in granular-ball construction result in numerous single-sample balls.

Motivated by the benefits of Information Bottleneck (IB) principle (Tishby et al., 1999; Hu et al., 2024a), researchers propose using granular balls as abstract representational carriers for data. These granular balls not only effectively compress raw data and filter out noise but also precisely preserve key discriminative features of data to prevent loss of core information during processing. Under concrete guidance of IB theory, generation and optimization of granular balls follow the core logic: maximize relevant information for each granular ball and minimize irrelevant interference. By dynamically adjusting coverage of a granular ball and composition of its internal samples, efficient information transfer and compact representation are achieved, which reduces computational burden of redundant data and strengthens ability of granular balls to capture local data structure.

In this paper, we propose a novel granular information bottleneck (GIB) method for DMC (shown in Fig. 1). The core idea of GIB is to introduce the concept of granular computing and employ granular balls as the basic representation units, thereby enhancing the model robustness and resistance to

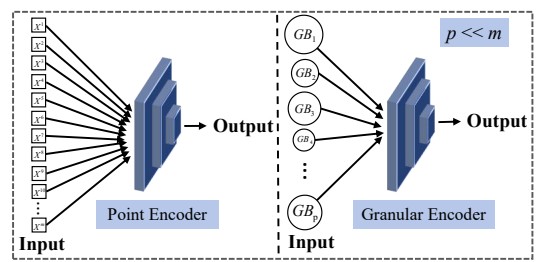 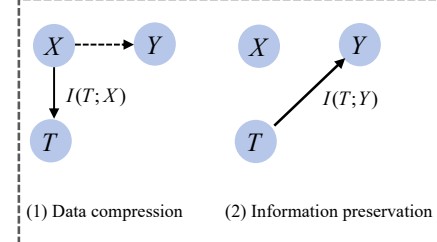

(a) Deep clustering methods comparison.          (b) Information Bottleneck model.

Figure 2: Deep clustering methods comparison and Information Bottleneck model. (a) The left figure is the existing sample-based clustering, where the inputs are multiple individual samples $\{X^i\}_{i=1}^m$. The right figure is granule-based clustering, which takes multiple granules $\{GB_i\}_{i=1}^p$ as input, with $p \ll m$ indicating a much smaller granule count than sample count. (b)(1) The dashed line indicates a joint distribution $p(X, Y)$ between $X$ and $Y$, while the solid line shows compression from $X$ to $T$. (2) The black solid line signifies that variable $T$ retains relevant information about $Y$.

noise. Specifically, at the lower level, the granular ball model adaptively generates multi-scale structures to mitigate intra-modal noise interference. At the higher level, we leverage granular balls as representation units to achieve cross-modal information preservation and compression. During compression, inter-modal discriminability is enhanced by minimizing the mutual information between modal features and cluster assignments. In preservation, the global shared representation guides the feature learning and clustering assignment of each modality, which can achieve cluster-level consistency while discovering feature-level commonalities. The main contributions are as follows:

- We propose a novel GIB method for deep multi-modal clustering, which introduces the granular structure to enhance noise resistance and improve model robustness. To our knowledge, it is the first method that introduces granular computing into deep multi-modal clustering, which introduces a new paradigm in multi-modal clustering.

- A novel dual-level information bottleneck constraint mechanism is designed, which performs at both the granular and sample levels to achieve collaborative optimization of feature representation and clustering assignment.

- A unified variational optimization framework is designed, which can efficiently solve the objective iteratively and drive the model to converge.

- We conduct extensive experiments which validate the superior accuracy, robustness of our model, demonstrating the effectiveness of the proposed granular representation and multi-modal integration strategy in achieving state-of-the-art clustering performance.

## 2 THE PROPOSED METHOD

### 2.1 PRIOR KNOWLEDGE

**Granular Computing.** Granular Computing proposed by Xia et al. (2024) treats the entire dataset as the coarsest granularity and performs splitting and refining of granular balls from coarse to fine to achieve a scalable and robust computing process. The core assumption is that data with similar distributions tend to form a granular ball, and adjacent granular balls further merge to form larger clustering. Specifically, these granular balls act as a means to cover and represent the data, thereby providing an accurate characterization of the sample space by functioning as input units. Fig. 2(a) illustrates this by comparing granule-based clustering methods with existing sample-based approaches. This approach offers two key advantages: reduced computational complexity due to the significantly smaller number of granular balls compared with individual samples, and increased robustness resulting from the coarse-grained representation. Therefore, granular balls enhance deep clustering efficiency and reliability in data representation and analysis.

**Information Bottleneck.** Information Bottleneck (IB) principle (Tishby et al., 1999; Hu et al., 2024a) is an information-theoretic data analysis method. Given a source variable $X$ and a target

$Y$, the goal of the information bottleneck is to find an optimal compressed representation $T$ for $X$ while maximizing the information retained about the target $Y$ (shown in Fig. 2(b)). The objective function of the IB method can be formulated as follows:

$$\mathcal{L}_{\min} = I(T; X) - \beta I(T; Y), \tag{1}$$

where $\mathbf{I}(\mathbf{T}; \mathbf{X})$ is the mutual information between the compressed representation $T$ and the source variable $X$, and the smaller this value, the greater the **information compression**. $\mathbf{I}(\mathbf{T}; \mathbf{Y})$ denotes the mutual information between $T$ and the target $Y$, which means that the larger this value, the greater the **information preservation**. The parameter $\beta$ balances compression and preservation.

In recent years, Information Bottleneck (IB) frameworks have increasingly been adopted within multi-modal clustering paradigms to facilitate the extraction of task-relevant information. While Federici et al. (2020) focuses on non-shared information, it neglects cluster-assignment consistency. Yan et al. (2023) jointly incorporates features and assignments but limits representational capacity by not fully exploiting clustering cues in global representations. Lou et al. (2025) introduce super deep contrastive information bottleneck, while incorporating hidden-layer features and dual contrastive objectives, is complex and hard to train. These sample-level methods thus undermine efficiency, robustness, and interpretability on high-dimensional, multi-modal data.

## 2.2 PROBLEM FORMULATION

Let the random variables $\{X^1, X^2, \ldots, X^m\}$ represent the observable data from $m$ modalities. For the $i$-th modality, the data is $X^i = \{x_1^i, x_2^i, \ldots, x_n^i\} \in \mathbb{R}^{n \times d^i}$, where $n$ is the sample count and $d^i$ is the feature dimension of the samples. The granular balls for modality $i$ are $\mathcal{G}^i = \{GB_1^i, GB_2^i, \ldots, GB_j^i\}$, where $GB_j^i$ is the $j$-th granular ball generated for this modality. $\{H^i\}_{i=1}^m$ represents the compressed feature representation from input $\{X^i\}_{i=1}^m$, while $H^s$ is the fused feature representation of the individual modalities. $\{Y^i\}_{i=1}^m$ denotes the local clustering assignment of $\{H^i\}_{i=1}^m$ obtained from the clustering model, and $Y^s$ represents the global clustering assignment for the fused feature $H^s$. $\mathcal{G}^s$ are the granular balls generated by feature fusion $H^s$. $\{C^i\}_{i=1}^m$ and $C^s$ denote granular-ball centers from individual modalities and fused features, respectively. $\theta$ denotes the pseudo-purity threshold and $n_{th}$ is the maximum sample threshold within a granular ball. Note that $k$ is the number of clusters in each modality.

The objective of the proposed GIB in this work is to eliminate multi-modal irrelevant information while effectively obtaining valuable and more discriminative features guided by the compression and preservation of granular information bottleneck.

## 2.3 PROPOSED OBJECTIVE FUNCTION

The overall objective of the GIB method proposed in this paper is achieved by minimizing the following loss function:

$$\mathcal{L}_{\text{Total}} = \mathcal{L}_{\text{Balls}} + \mathcal{L}_{\text{Samples}} + \mathcal{L}_{\text{DDC}}. \tag{2}$$

In this hierarchical loss function, $\mathcal{L}_{\text{Balls}}$ focuses on granular feature learning, while $\mathcal{L}_{\text{Samples}}$ denotes sample clustering alignment. Both utilize the information bottleneck principle to retain essential task-relevant information and reduce noise. And the $\mathcal{L}_{\text{DDC}}$ represents the clustering module. Jointly optimizing the three components enables the GIB to achieve satisfactory clustering performance.

## 2.4 GRANULAR FEATURE LEARNING WITH $\mathcal{L}_{\text{BALLS}}$

### 2.4.1 GRANULAR BALL REPRESENTATION

The center $c_j$ of the granular ball $GB_j = \{x_j\}_{j=1}^{n_j}$ is represented by the mean of all data points within it. The radius $r_j$ is represented by the average distance from all data points to its center. The $c_j$ and $r_j$ of $GB_j$ are defined as:

$$c_j = \frac{1}{n_j} \sum_{i=1}^{n_j} x_i, \quad r_j = \frac{1}{n_j} \sum_{i=1}^{n_j} \|x_i - c_j\|_2, \tag{3}$$

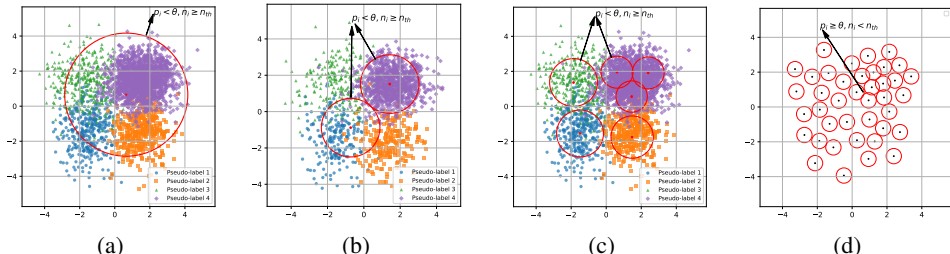

(a)      (b)      (c)      (d)

Figure 3: Process of granular ball splitting: purity threshold is set to $\theta$, and the granular ball quantity threshold is $n_{th}$. The three colors of sample points in the figure represent the pseudo-labels assigned by weighted K-Means. (a) The initial granular-ball, the entire dataset can be seen as a granular-ball to participate in subsequent iterations; (b) Granular-balls generated in the first iteration; (c) Intermediate result; (d) Granular-balls extracted.

where $\|x_j - c_j\|$ denotes the Euclidean distance from $x_j$ to $c_j$, and $n_j$ is the number of samples in $GB_j$. We set a threshold $n_{th}$ for the number of samples in each granular ball to achieve a coarse-grained representation, and also establish a granular ball purity threshold $\theta$. Fig. 3 illustrates the method for generating granular balls. Initially, the entire dataset is treated as a single large granular ball (shown in Fig. 3(a)). The algorithm calculates the purity of each granular ball and the number of samples it contains. If the purity of the granular ball is greater than the threshold $\theta$, and the number of samples within the granular ball is less than the threshold $n_{th}$, then the granular ball is retained. Otherwise, the granular ball continuously splits (shown in Fig. 3(b) and Fig. 3(c)) until these conditions are met or the iteration stopping criteria are reached. Finally, granular balls that meet the conditions are extracted (shown in Fig. 3(d)). The different colored sample points in the figure represent the pseudo-labels assigned by the weighted K-Means algorithm. Given that K-Means is sensitive to initial centers, the resulting pseudo-labels are highly dependent on the selection of those starting points. The purity calculation is as follows:

$$\text{Pseudo-Purity}(GB_j) = \frac{\max_{1 \leq i \leq k}\left(\text{Count}(i)\right)}{|n_j|}, \tag{4}$$

where $k$ is the number of clusters defined for pseudo-labeling. $\text{Count}(i)$ is the number of samples in the granular ball $GB_j$ that belong to the $i$-th pseudo-label cluster. High pseudo-purity signifies good quality, indicating a granular ball is dominated by samples of a single class. Conversely, low pseudo-purity reflects uneven distribution and poorer quality. An excessively high pseudo-purity threshold can lead to numerous small granular balls, hindering the capture of data distribution structure. However, too low a threshold may result in insufficient splitting, retaining redundant information and noise. Therefore, selecting a suitable purity threshold $\theta$ is important.

### 2.4.2 GRANULAR BALL FEATURE LEARNING LOSS

Using granular balls as processing units, we apply the information bottleneck principle at the feature level to compress yet preserve information. Each ball is represented by its center $c$ for mutual-information computation. The granular level feature learning objective is:

$$\mathcal{L}_{\text{Balls}} = \alpha \sum_{i=1}^{m} \sum_{j=i+1}^{m} I(C^i, C^j) - \beta \sum_{i=1}^{m} I(C^i, C^s), \tag{5}$$

where $I(\cdot, \cdot)$ represents the mutual information between two variables. $C^i$ and $C^j$ are the sets of granular ball centers generated from modalities $i$ and $j$ respectively. The fusion mechanism aggregates specific modal features into a shared representation $H^s$, as further described in the appendix A.6. $C^s$ denotes the set of granular ball centers generated from the shared representation $H^s$. $\alpha, \beta \in (0, 1)$ are the balance parameters trading off the information compression and preservation.

The first term in Eq. 5 is the mutual information between the private information of each modality, which reduces redundant information and enhances the complementarity among the modalities. The

second term is the mutual information between the global representation and each specific modality, which enhances the ability to capture effective information from each modality, thereby forming a richer and more discriminative globally shared representation. By minimizing this objective function $\mathcal{L}_{\text{Balls}}$, the GIB can learn more optimal feature representation.

## 2.5 SAMPLE CLUSTERING ALIGNMENT WITH $\mathcal{L}_{\text{SAMPLES}}$

We cluster the granular balls and then map the results back to the samples, as shown in Fig. 1. The objective function for sample clustering alignment is as follows:

$$\mathcal{L}_{\text{Samples}} = \alpha \left\{ \sum_{i=1}^{m} I(X^i, H^i) + \sum_{i=1}^{m} \sum_{j=i+1}^{m} I(Y^i, Y^j) \right\} - \beta \sum_{i=1}^{m} I(Y^i, Y^s), \qquad (6)$$

where $\alpha, \beta \in (0, 1)$ are balancing parameters controlling information compression and preservation. These parameters match the granular-level settings, ensuring balanced contributions from both hierarchical levels. The mutual information between the original features and the private information $I(X^i, H^i)$ is used to remove invalid information, ensuring that only critical features of each modality are retained for downstream tasks. $I(Y^i, Y^j)$ measures the correlation between clustering assignments of all modalities. And the $I(Y^i, Y^s)$ calculates the mutual information between the clustering assignment of each modality and the global clustering assignment to achieve the consistency of cross-modal clustering assignments.

## 2.6 DATA CLUSTERING MODULE WITH $\mathcal{L}_{\text{DDC}}$

In this paper, we present the Deep Divergence-based Clustering (DDC) loss function, which optimizes clustering performance through three key constraints: the Cauchy-Schwartz divergence measures the difference between cluster centers and the overall data distribution; orthogonality is enforced among clustering vectors; and a spatial morphology constraint prevents trivial solutions, directing the learning process toward meaningful feature distributions. The DDC loss function is:

$$\mathcal{L}_{\text{DDC}} = \frac{1}{k} \sum_{i=1}^{k-1} \sum_{j>i} \frac{\mu_i^T E_{\mu_j}}{\sqrt{\mu_i^T E_{\mu_i \mu_j} \mu_j^T E_{\mu_j}}} + \text{triu}(A^T A) + \frac{1}{k} \sum_{i=1}^{k-1} \sum_{j>i} \frac{\gamma_i^T E_{\gamma_j}}{\sqrt{\gamma_i^T E_{\gamma_i \gamma_j} \gamma_j^T E_{\gamma_j}}}, \qquad (7)$$

where $k$ is the total number of clusters, and $E$ represents a matrix derived using a Gaussian kernel function. The term $\mu_i$ denotes the $i$-th column vector in the clustering result matrix $A$. Additionally, $\gamma_i$ is determined by the $i$-th column vector of the matrix $U_{ab} = \exp(-\|\alpha_a - e_b\|^2)$, where $e_b$ represents the $b$-th vertex of the simplex. The expression $\text{triu}(A^T A)$ represents the sum of the elements in the upper triangular part of the matrix $A^T A$, excluding the diagonal elements.

## 2.7 OPTIMIZATION

We propose a variational optimization method to solve the objective function by approximating mutual information as a trainable loss function. This mutual information involves two variables (the first modality as an example):

$$\begin{aligned} I(X^1; H^1) &= \int_{h^1} \int_{x^1} p(x^1, h^1) \log \frac{p(x^1, h^1)}{p(x^1)p(h^1)} \, dx^1 dh^1 \\ &= \int_{h^1} \int_{x^1} p(x^1, h^1) \log \frac{p(x^1 \mid h^1)}{p(x^1)} \, dx^1 dh^1. \end{aligned} \qquad (8)$$

We approximate the posterior distribution $P(x^1, h^1)$ using the variational distribution $q(x^1)$ and use the Kullback-Leibler (KL) divergence measure to constrain their discrepancy. Here, we introduce a key theorem of KL divergence regarding posterior inference.

**Theorem 1 (Posterior Approximation via KL Divergence)** *Since KL divergence is non-negative, by minimizing $KL(q(x^1)\|p(x^1))$, the approximate posterior $q(x^1)$ approaches the true posterior $p(x^1)$.* *Proof.* See Appendix A.2.

Based on Theorem 1, we can now rewrite the mutual information $I(X^1; H^1)$ as follows:

$$I(X^1; H^1) = \iint p(x^1, h^1) \log \frac{p(x^1 \mid h^1)}{p(x^1)} < \iint p(x^1, h^1) \log \frac{p(x^1 \mid h^1)}{q(x^1)}. \tag{9}$$

Since Theorem 1 establishes the non-negativity of KL divergence and controls the approximate posterior, we can approximate the mutual information $I(X^1; H^1)$ using the log-likelihood ratio decomposition and variational representation, leading to the expression in Theorem 2.

**Theorem 2 (Variational Mutual Information Approximation)** *Assuming the conditional distribution $p(x^1 \mid h^1)$ is Gaussian with learnable mean $\mu$ and variance $\sigma$ (via a variational IB encoder), the mutual information $I(X^1; H^1)$ admits the approximation:*

$$I(X^1; H^1) \approx \frac{1}{M} \sum_{i=1}^{M} \mathbb{E}_{\theta_i} \left\{ KL \left[ p(x^1 \mid h^1) \big\| q(x^1) \right] \right\}, \quad \sum_{i=1}^{M} q(x^1) = \frac{n}{k},$$

*where $\theta_i \sim \mathcal{N}(0, 1)$ and $q(x^1)$ is uniformly distributed to enforce balanced cluster assignments.*
*Proof.* See Appendix A.3.

Building upon the variational approximation of mutual information established in Theorem 2, Proposition 1 focuses on its calculation and optimization for information compression and preservation.

**Proposition 1 (Calculation of Mutual Information)** *By computing the joint probability $p(C^i, C^j)$ and the marginal probabilities $p(C^i)$ and $p(C^j)$ of the latent representations, mutual information can be computed as follows:*

$$I(C^i, C^j) = \sum_{i=1}^{m} \sum_{j=i+1}^{m} \mathbb{1}_{i \neq j} p(C^i, C^j) \log \left( \frac{p(C^i, C^j)}{p(C^i) p(C^j)} \right). \tag{10}$$

*Proof.* See Appendix A.4. Similarly, according to Proposition 1, $I(C^i, C^s)$, $I(Y^i, Y^s)$, as well as $I(Y^i, Y^j)$ can be computed. The details can be found in Algorithm 1 (in the appendix A.5).

## 3 EXPERIMENTS

### 3.1 EXPERIMENTAL SETUP

We briefly introduce the experimental setup here, including the experimental datasets, evaluation metrics, model selection, and comparison methods.

**Datasets and Backbones.** We conducted experiments on five publicly available and well-known datasets: Caltech-2V, Caltech-3V, WVU, IAPR, and MIRFlickr. These datasets have different modalities and sample sizes. For a more detailed description of the dataset, please refer to appendix A.7. The GIB model uses a unified MLP network architecture across all datasets, consisting of three fully connected layers with ReLU activation functions. The output dimensions of the layers are 512, 512, and 256, respectively.

**Evaluation Metrics.** We use Accuracy (ACC) and Normalized Mutual Information (NMI) to evaluate the final clustering performance. ACC is used to quantify the consistency between clustering results and true labels, while NMI measures the degree of information shared between clustering results. Higher values for both metrics indicate better clustering performance.

**Implementation Details.** We implemented GIB and other methods for comparison on a Windows 10 system equipped with a 24 GB NVIDIA RTX-4090 GPU, using the PyTorch 1.13.0 platform (Python version 3.9). We ran the model 20 times. In each run, the training process converged after 100 epochs, and we carefully selected the model with the highest accuracy and lowest loss. The batch size was set to 256, and Adam was chosen as the optimizer with a learning rate of 0.0001. We fixed the pseudo-purity threshold and the maximum sample number threshold within the granular ball to 0.9 and 1.0, respectively.

Table 1: Clustering performance with ACC and NMI on various kinds of datasets (the bold and underlined values in the table represent the best and second-best results respectively).

| Methods | Caltech-2V | | Caltech-3V | | WVU | | IAPR | | MIRFlickr | |
| --- | --- | --- | --- | --- | --- | --- | --- | --- | --- | --- |
| | ACC | NMI | ACC | NMI | ACC | NMI | ACC | NMI | ACC | NMI |
| KM | 41.6 | 30.5 | 46.3 | 31.3 | 30.8 | 37.2 | 38.9 | 17.2 | 40.9 | 22.5 |
| Ncuts (TPAMI'00) | 39.9 | 31.2 | 42.6 | 25.4 | 55.9 | 41.9 | 41.9 | 18.9 | 48.4 | 26.1 |
| AmKM | 44.6 | 35.2 | 46.9 | 31.5 | 27.9 | 25.1 | 40.4 | 17.0 | 41.0 | 21.6 |
| AmNcuts (TPAMI'00) | 42.8 | 52.2 | 43.7 | 25.5 | 58.3 | 55.0 | 42.2 | 18.9 | 48.2 | 26.2 |
| CoregMVSC (NIPS'11) | 49.2 | 39.6 | 54.4 | 45.3 | 36.5 | 55.8 | 35.1 | 18.4 | 41.0 | 26.8 |
| RMKMC (IJCAI'13) | 51.4 | 33.5 | 59.5 | 49.4 | 46.0 | 53.3 | 36.4 | 15.9 | 42.3 | 23.4 |
| SwMC (IJCAI'17) | 49.9 | 37.1 | 54.8 | 43.3 | 41.8 | 10.1 | 30.2 | 23.1 | 34.3 | 34.5 |
| ONMSC (AAAI'20) | 34.2 | 26.6 | 30.2 | 23.1 | 28.9 | 27.9 | 21.6 | 11.1 | 30.6 | 16.4 |
| EAMC(CVPR'20) | 41.9 | 25.6 | 38.9 | 21.4 | 26.9 | 15.2 | 37.1 | 16.4 | 30.5 | 9.1 |
| DEMVC(InfoSci'21) | 39.4 | 22.2 | 38.7 | 27.0 | 49.1 | 50.9 | 30.1 | 13.8 | 44.8 | 25.2 |
| SiMVC (CVPR'21) | 50.8 | 47.1 | 56.9 | 50.4 | 46.6 | 45.2 | 42.7 | 18.5 | 45.6 | 26.3 |
| CoMVC (CVPR'21) | 46.6 | 42.6 | 54.1 | 50.4 | 42.3 | 44.4 | 46.7 | 21.5 | 49.3 | 30.6 |
| MFLVC (CVPR'22) | 60.6 | 52.8 | 63.1 | 56.6 | 58.2 | 51.3 | 47.3 | 22.6 | 53.8 | 32.8 |
| SPDMC (TNNLS'23) | 64.4 | 50.6 | 70.1 | 63.0 | 32.9 | 31.3 | 33.3 | 17.1 | 47.5 | 30.3 |
| DealMVC (ACM MM'23) | 60.0 | 50.0 | 59.5 | 56.8 | 55.2 | 56.4 | 35.0 | 10.8 | 49.3 | 32.1 |
| ICMVC (AAAI'24) | 39.0 | 25.0 | 53.2 | 40.3 | 38.3 | 39.0 | 37.1 | 16.8 | 43.5 | 24.4 |
| DIVIDE (AAAI'24) | 64.1 | 52.9 | 67.8 | 56.2 | 49.9 | 50.0 | 45.6 | 23.0 | 52.3 | 33.5 |
| PDMC-RCL(TIP'25) | 62.5 | 52.4 | 69.7 | 58.4 | 58.0 | 49.1 | 45.7 | 22.4 | 52.7 | 33.0 |
| CCMVC(TNNLS'25) | 58.5 | 49.7 | 59.5 | 54.0 | 47.4 | 47.3 | 38.4 | 20.8 | 52.3 | 33.4 |
| **GIB** | **69.7** | **57.2** | **74.3** | **64.0** | **64.0** | **56.8** | **50.6** | **25.4** | **56.2** | **35.8** |
| **Ours vs Best Compared** | **5.3↑** | **4.3↑** | **4.2↑** | **1.0↑** | **5.7↑** | **0.4↑** | **3.3↑** | **2.8↑** | **2.4↑** | **2.3↑** |

**Compared Methods.** We compare the proposed method with the following models: **(a) Single-modal clustering methods:** Perform clustering on each individual modality for multi-modal data. Typical algorithms include K-Means (KM) and Normalized Cuts (Ncuts). **(b) Full-modal clustering methods:** Connect all modalities and then apply single-modality clustering methods. Representative algorithms include AmKM (All-modal K-Means) and AmNcuts (All-modal NCuts). **(c) Traditional multi-modal clustering methods:** RMKMC Cai et al. (2013), ONMSC Zhou & Shen (2020), CoregMVSC Kumar et al. (2011) and SwMC Nie et al. (2017). **(d) Deep multi-modal clustering methods:** EAMC Zhou & Shen (2020), DEMVC (Xu et al., 2021), SiMVC and CoMVC (Trosten et al., 2021), SPDMC Chen et al. (2023), MFLVC (Xu et al., 2022), DealMVC (Yang et al., 2023), ICMVC Chao et al. (2024), DIVIDE Lu et al. (2024), PDMC-RCL Lou et al. (2025) and CCMVC Shi et al. (2025). For detailed descriptions of these methods, please refer to appendix A.8.

## 3.2 EXPERIMENTAL RESULTS

We compare our method with 19 state-of-the-art multi-modal clustering methods and present the clustering results on the involved multi-modal datasets in Table 1. To further illustrate the effectiveness of our method, we provide intuitive clustering visualizations in the appendix A.9.

**Comparison on the Overall Dataset.** GIB achieved substantial improvements across all datasets, as measured by the ACC and NMI metrics. Taking the Caltech-2V dataset as an example, our method improved by 5.3% in ACC and 4.3% in NMI compared to the second-best method (DIVIDE). This indicates that the proposed GIB method has significant advantages in deep multi-modal clustering.

**Comparison on Small-Scale Datasets.** In the comparison on the WVU dataset, GIB outperformed AmNcuts by 5.7% in ACC, demonstrating its ability to effectively capture key features in small-scale datasets by correlating granular balls and leveraging feature compression and preservation.

**Comparison on Large-Scale Datasets.** In the comparison on the IAPR dataset, GIB achieves 2.5% higher ACC than the second-best method (MFLVC). This is because direct processing of massive samples by traditional clustering incurs high computational costs and local optima issues. In contrast, GIB replaces samples with granular balls, improving computational efficiency and preserving inter-cluster differences for large-scale clustering.

Table 2: Ablation experiments on multi-modal datasets.

| Methods | Caltech-2V | | Caltech-3V | | WVU | | IAPR | | MIRFlickr | |
|---|---|---|---|---|---|---|---|---|---|---|
| | ACC | NMI | ACC | NMI | ACC | NMI | ACC | NMI | ACC | NMI |
| (1) $\mathcal{L}_{DDC}$ | 63.5 | 50.1 | 63.9 | 52.9 | 45.6 | 38.8 | 43.4 | 21.4 | 46.4 | 26.0 |
| (2) $\mathcal{L}_{DDC} + \mathcal{L}_{Balls}$ | 64.6 | 55.2 | 71.5 | 64.4 | 54.4 | 49.8 | 48.2 | 24.4 | 48.6 | 26.3 |
| (3) $\mathcal{L}_{DDC} + \mathcal{L}_{Samples}$ | 65.7 | 54.1 | 69.1 | 60.5 | 59.6 | 54.8 | 44.1 | 23.0 | 47.9 | 25.8 |
| (4) $GIB$ | **69.7** | **57.2** | **74.3** | **64.0** | **64.0** | **56.8** | **50.6** | **25.4** | **56.2** | **35.8** |

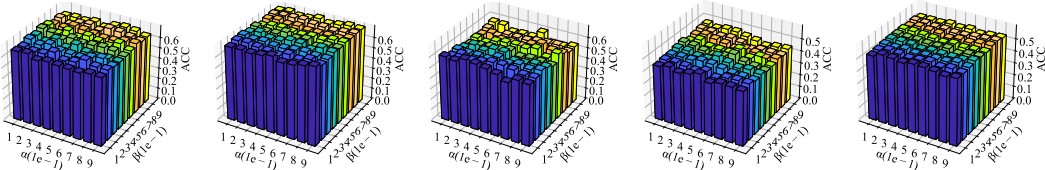

Figure 4: Parameter analysis of GIB on multi-modal datasets.

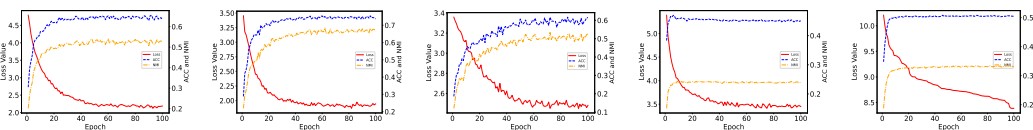

Figure 5: Convergence analysis of GIB on multi-modal datasets.

### 3.3 COMPREHENSIVE EVALUATION

**Ablation Study.** We conducted ablation experiments, and the results are shown in Table 2. When only the DDC clustering module is retained, the clustering performance is at its lowest level. Combining either the $\mathcal{L}_{Samples}$ module or the $\mathcal{L}_{Balls}$ module with the DDC module leads to performance improvements. The best results are achieved when all three modules are integrated. In summary, the experiments verify that the $\mathcal{L}_{Samples}$ and $\mathcal{L}_{Balls}$ modules synergistically enhance clustering performance, fully demonstrating the effectiveness of each module.

**Parameter Analysis.** To balance the compression and preservation processes, we set the same trade-off parameters $\alpha$ and $\beta$ for $\mathcal{L}_{Samples}$ and $\mathcal{L}_{Balls}$. For these two parameters, we used a grid search method for tuning, adjusting their values from 0 to 1 with a step size of 0.1. The results are shown in Fig. 4. Under most parameter settings, the clustering performance for each dataset tends to be consistent, indicating that the proposed method is not very sensitive to parameter changes.

**Convergence Analysis.** To evaluate the convergence of the proposed method, we present the changes of the overall loss function, ACC, and NMI over epochs. As shown in Fig. 5, the loss function decreases rapidly at the beginning and stabilizes around 100 epochs. Meanwhile, ACC and NMI increase simultaneously and converge to stable values. Both indicate that our method has satisfactory convergence properties.

## 4 CONCLUSION

This paper innovatively proposes the GIB multi-modal clustering method. By representing the finest-grained samples with large-scale granular balls, GIB can effectively eliminate noise within modalities and improve computational efficiency. GIB constrains feature representation and clustering assignment through information bottlenecks at both the granular and the sample levels, thereby learning more compact and discriminative representations while suppressing interference from irrelevant information. However, multi-modal data often contain missing information, and this incompleteness can prevent models from fully leveraging the complementary information across modalities, thereby degrading clustering performance. In the future, we plan to extend the proposed approach to handle incomplete multi-modal data.

ETHICS STATEMENT

This paper does not involve any potential ethics issues.

REPRODUCIBILITY STATEMENT

We have taken significant measures to guarantee the reproducibility of our research. The primary document outlines the proposed methodology, key algorithms, and evaluation criteria in detail. Additionally, the supplementary materials include comprehensive proofs of the theoretical claims and thorough derivations of the main results. For our experiments, we utilized publicly available datasets and provided an in-depth description of the experimental setup. To support reproducibility, the full source code and implementation specifics will be made publicly accessible upon the acceptance of this manuscript.

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

## A  APPENDIX

In the supplemental material:

- **A.1**: We discuss the use of the large language mode (LLM).
- **A.2**: We provide a detailed proof of Theorem 1.
- **A.3**: We provide a detailed proof of Theorem 2.
- **A.4**: We provide a detailed proof of Proposition 1.
- **A.5**: We describe details in Algorithm 1 to clearly present the proposed GIB framework.
- **A.6**: We describe the adopted modality fusion method.
- **A.7**: We give detailed descriptions of the datasets.
- **A.8**: We give a detailed description of the comparison methods.
- **A.9**: We conducted a detailed visual analysis of relevant datasets.

### A.1  LLM USAGE IN RESEARCH AND ANALYSIS

In this paper, we do not use any large language model.

### A.2  PROOF OF THEOREM 1

In the variational optimization carried out in this study, we use the Kullback–Leibler (KL) divergence to constrain the relationship between the approximate posterior distribution $q(x^1)$ and the true posterior distribution $p(x^1)$. The theorem below states the key properties of the KL divergence and its role in approximating posterior distributions.

*proof.*

Let $p(x^1)$ be a probability distribution and $q(x^1)$ be another probability distribution, with $q(x^1) > 0$ for all $x^1$. The Kullback–Leibler divergence $\mathrm{KL}(p(x^1)||q(x^1))$ is defined as:

$$\mathrm{KL}(p(x^1) \parallel q(x^1)) = \int p(x^1) \log \frac{p(x^1)}{q(x^1)} \, dx^1. \tag{11}$$

By the inequality of Jensen, or directly from basic properties of probability, the KL divergence is non-negative:

$$\mathrm{KL}\big(p(x^1) \parallel q(x^1)\big) \geq 0. \tag{12}$$

The equality holds if and only if $p(x^1) = q(x^1)$ almost everywhere. For any $p(x^1)$ and $q(x^1)$ that satisfy the appropriate domain and positivity conditions, we have:

$$\int p(x^1) \log \frac{p(x^1)}{q(x^1)} \, dx^1 \geq 0. \tag{13}$$

As a further transformation, the fractional expression within the logarithm is split into the difference of two logarithms:

$$\log \frac{p(x^1)}{q(x^1)} = \log p(x^1) - \log q(x^1). \tag{14}$$

Substituting into the above equation, we get:

$$\int p(x^1) \log p(x^1) \, dx^1 - \int p(x^1) \log q(x^1) \, dx^1 \geq 0. \tag{15}$$

After rearranging terms, we obtain:

$$\int p(x^1) \log p(x^1) \, dx^1 \geq \int p(x^1) \log q(x^1) \, dx^1. \tag{16}$$

This means that by minimizing $\mathrm{KL}(p(x^1)||q(x^1))$ we can drive the approximate posterior $q(x^1)$ to approach the true posterior $p(x^1)$. This is because as the KL divergence decreases, the difference between $q(x^1)$ and $p(x^1)$ is continuously reduced, and eventually $q(x^1)$ can approximate $p(x^1)$ well.

### A.3 PROOF OF THEOREM 2

*proof.*

By definition of mutual information, we have:

$$I(X^1; H^1) = \iint p(x^1, h^1) \log \frac{p(x^1 \mid h^1)}{p(x^1)} \, dx^1 \, dh^1. \tag{17}$$

Here, $p(x^1, h^1)$ is the joint probability density of $x^1$ and $h^1$, $p(x^1 \mid h^1)$ is the conditional probability density, and $p(x^1)$ is the marginal probability density of $x^1$.

Given the relationship between $p(x^1)$ and the distribution $q(x^1)$ used for optimization, it follows that:

$$I(X^1; H^1) < \iint p(x^1, h^1) \log \frac{p(x^1 \mid h^1)}{q(x^1)} \, dx^1 \, dh^1. \tag{18}$$

Since the joint probability can be expressed as:

$$p(x^1, h^1) = p(h^1)p(x^1 \mid h^1), \tag{19}$$

substitution into Eq. 18 yields:

$$I(X^1; H^1) < \sum_{i=1}^{m} \iint p(h^1)p(x^1 \mid h^1) \log \frac{p(x^1 \mid h^1)}{q(x^1)} \, dx^1 \, dh^1. \tag{20}$$

To remove redundant terms, Monte Carlo sampling Von Ahn & Dabbish (2004) is used for approximation and replacement of $p(h^1)$ to obtain a more accurate estimation. After derivation, the mutual information can be further expressed as:

$$I(X^1; H^1) < \sum_{i=1}^{m} \int p(x^1 \mid h^1) \log \frac{p(x^1 \mid h^1)}{q(x^1)}. \tag{21}$$

The role of Monte Carlo sampling is to approximate integrals by drawing samples, avoiding the difficulty of computing high-dimensional integrals directly and thereby improving computational feasibility and accuracy.

Assume that $p(x^1 \mid h^1)$ follows a Gaussian distribution, whose mean $\mu$ and variance $\sigma$ can be learned by the variational IB encoder. To simplify calculations, $h^1$ is reparameterized as $h^1 = \mu(x^1) + \sigma(x^1) \cdot \theta$, where $\theta$ represents the standard normal distribution. At this time, the mutual information can be expressed as:

$$I(X^1; H^1) < \sum_{i=1}^{m} \left\{ \mathbb{E}_{\theta_i} \log \frac{p(x^1 \mid h^1)}{q(x^1)} \right\} < \sum_{i=1}^{m} \mathbb{E}_{\theta_i} \left\{ \text{KL} \left[ p(x^1 \mid h^1) \parallel q(x^1) \right] \right\}. \tag{22}$$

To ensure that data samples are evenly divided into all categories, a constraint is set on $q(x^1)$ based on the uniform distribution: $\sum_{i=1}^{M} q(x^1) = \frac{M}{k}$ (where $M$ is the number of data instances, and $k$ is the number of clusters). Combining with the number of data instances $M$, the final mutual information is approximated as:

$$I(X^1; H^1) \approx \frac{1}{M} \sum_{i=1}^{M} \mathbb{E}_{\theta_i} \left\{ \text{KL} \left[ p(x^1 \mid h^1) \parallel q(x^1) \right] \right\}, \quad \sum_{i=1}^{M} q(x^1) = \frac{M}{k}, \tag{23}$$

which completes the proof.

### A.4 PROOF OF PROPOSITION 1

*proof.*

To calculate $I(C^i, C^j)$, we first obtain the joint probability distribution $p(C^i, C^j)$ by applying dimension expansion, element-wise multiplication, and summation over the sample dimension, as described in Eq. 24:

$$p(C^i, C^j) = \sum_{n=1}^{b_n} C_n^i \times (C_n^j)^T. \tag{24}$$

Here, $b_n$ represents the batch size. The joint probability matrix is symmetric and normalized, as shown in Eq. 25:

$$p(C^i, C^j) = \frac{1}{2} \left( p(C^i, C^j) + p(C^j, C^i) \right), \quad p(C^i, C^j) = \frac{p_{ij}}{\sum_{i,j} p_{ij}}. \tag{25}$$

The first formula guarantees symmetry, and the second formula normalizes all probabilities to sum to 1. Next, we calculate the marginal probabilities $p(C^i)$ and $p(C^j)$ of $I(C^i, C^j)$. The formula for calculating mutual information $I(C^i, C^j)$ is as follows:

$$I(C^i, C^j) = \sum_{i=1}^{m} \sum_{j=i+1}^{m} \Vdash_{i \neq j} p(C^i, C^j) \log \left( \frac{p(C^i, C^j)}{p(C^i)p(C^j)} \right). \tag{26}$$

Here, the summation $\sum_{\forall k \neq i}$ denotes summing over all indices $k$ different from $i$.

Similarly, $I(C^i, C^s)$ and $I(Y^i, Y^s)$ are obtained by calculating their respective joint probabilities and marginal probabilities.

## A.5 Algorithmic Description of the Deep Granular Information Bottleneck for Multi-Modal Clustering

---
**Algorithm 1** GIB Algorithm

---
1: **Input**: Multi-modal dataset $\{X^i\}_{i=1}^{m}$, number of clusters $k$, hyper-parameters $\alpha, \beta$, learning rate $\gamma$, purity threshold $\theta$, and granular ball quantity threshold $n_{\text{th}}$.
2: **Output**: The clustering result.
3: Initialize the neural network parameters.
4: **while** not converge **do**
5: Extract modal-specific representations $\{H^i\}_{i=1}^{m}$ by sharing modal-specific encoders.
6: Generate the granular balls $\{\mathcal{G}^i\}_{i=1}^{m}$ of each modality in the latent feature space.
7: Calculate the granular-level feature learning loss function using Eq. 5.
8: Calculate the sample-level clustering alignment loss function using Eq. 6.
9: Calculate the DDC loss using Eq. 7.
10: Jointly optimize the overall loss function by Eq. 2.
11: **end while**
12: **return** obtaining the final clustering result.

---

## A.6 Multi-modal Fusion

The shared features among different modalities can be used to learn feature correlations. We did not adopt the widely used current approach that automatically assigns weights to each view via an attention mechanism. Instead, we propose a simplified strategy to learn view weights directly from the clustering objective itself. The reasons for choosing this strategy are as follows: introducing an additional attention-learning module could render the entire network quite bulky, thereby increasing the runtime and memory overhead. By contrast, these view weights can be directly updated by optimizing the clustering objective during joint training. The definition is as follows:

$$S = \sum_{i=1}^{m} w^i H^i. \tag{27}$$

Specifically, we first initialized the weight of each modality to $\frac{1}{m}$ to ensure balance in early contributions, and then dynamically updated the specific weights for each modality through backpropagation, while ensuring that the sum of the weights remains ($\sum_{i=1}^{m} w^i = 1$).

## A.7 DATASETS DETAILS

We describe the datasets used in the experiments in detail and summarize the datasets in Table 3.

Table 3: Description of five multi-modal datasets.

| Dataset | Modalities | Samples | Clusters |
|---------|-----------|---------|----------|
| Caltech-2V | 2 | 1400 | 7 |
| Caltech-3V | 3 | 1400 | 7 |
| WVU | 4 | 650 | 10 |
| IAPR | 2 | 7855 | 6 |
| MIRFlickr | 4 | 12154 | 6 |

- **Caltech-2V** Fei-Fei et al. (2004) contains 1,440 images covering 7 object categories. It includes two types of features: Wavelet moments Shen & Ip (1999) and CENsus TRansform hISTogram (CENTRIST) Wu & Rehg (2010).

- **Caltech-3V** contains the same categories and images as Caltech-2V but introduces an additional feature.

- **WVU** Ramagiri et al. (2011) dataset is derived from action data and includes four different modalities. All videos undergo feature detection and description using the Harris3-D detector and Spatio-Temporal Interest Points (STIP) with HoG/HoF descriptors.

- **IAPR** Grubinger et al. (2006) contains 7,855 images and their corresponding textual descriptions, forming two modalities and covering six categories.

- **MIRFlickr** Huiskes & Lew (2008) contains 12,154 images, which are divided into 6 different categories.

## A.8 COMPARISON METHOD DETAILS

To ensure a fair comparison, we downloaded the source code of the competing methods from the authors' websites and ran them according to the experimental settings and parameter-tuning procedures described in each paper.

- **K-Means**: Partition data points into $K$ clusters to maximize similarity within clusters and minimize dissimilarity between clusters. The algorithm iteratively updates cluster centers and assigns points until convergence.

- **Ncuts (Normalized Cuts)**: It is a graph-based clustering method that achieves data grouping by minimizing the normalized cut cost of the graph.

- **AmKM**: An adaptive clustering method dynamically adjusts the cluster centers and their number to better fit the distribution characteristics of the data.

- **AmNcuts**: It actively groups nodes into high-quality clusters for knowledge graphs and multi-modal data using the Normalized Cut (N-cut) principle, aiming to minimize cross-cluster edge weights and maximize intra-cluster connectivity.

- **CoregMVSC** Kumar et al. (2011): Co-regularize clustering hypotheses to achieve consistent cluster assignments across modalities.

- **RMKMC** Cai et al. (2013): Proposes a robust and scalable multi-view clustering method that integrates heterogeneous representations of large-scale data for unsupervised clustering. This method overcomes the limitations of graph-based spectral clustering on large-scale datasets.

- **SwMC** Nie et al. (2017): Introduces a Laplacian rank-constrained graph to learn view weights and directly assigns cluster labels to each data point without any post-processing.

- **EAMC** Zhou & Shen (2020): Learns cross-modal features via a shared encoder and introduces a reconstruction constraint to strengthen feature-level contrastive learning (EnFeaCL), addressing existing methods' shortcomings with noisy data and local diversity.

- **DEMVC** Xu et al. (2021): It uses deep autoencoders to learn embeddings for each modality, and jointly optimizes feature representations and cluster assignments during co-training while accounting for modality consistency and complementarity.
- **CoMVC** Trosten et al. (2021): Proposes a deep multimodal clustering baseline with unaligned representations that can match or surpass the SOTA even without alignment, and uses contrastive learning to enable selective alignment while preserving each modality's priority.

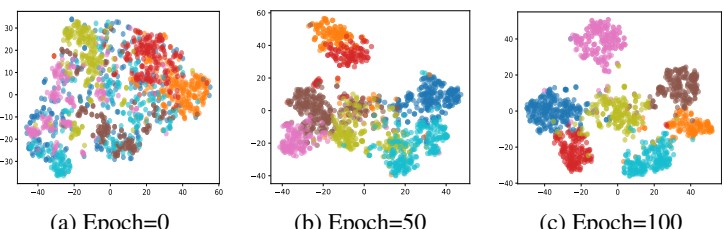

(a) Epoch=0     (b) Epoch=50     (c) Epoch=100

Figure 6: Visualization on the Caltech2V dataset.

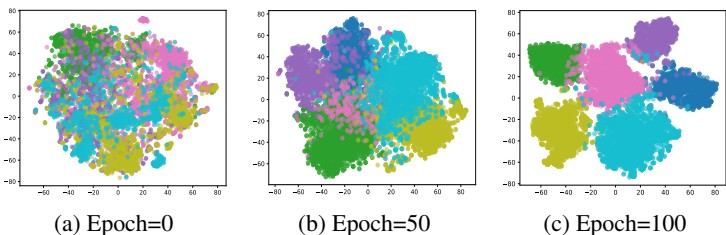

(a) Epoch=0     (b) Epoch=50     (c) Epoch=100

Figure 7: Visualization on the IAPR dataset.

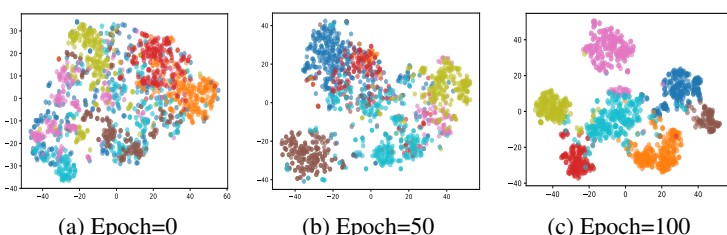

(a) Epoch=0     (b) Epoch=50     (c) Epoch=100

Figure 8: Visualization on the Caltech3V dataset.

- **MFLVC** Xu et al. (2022): Addresses the conflict between learning consistent semantics and reconstructing modality-specific information by learning low-level and high-level features independently in separate feature spaces, significantly improving clustering performance.
- **SPDMC** Chen et al. (2023): It constrains sample-pair relationships with prior knowledge through a unified regularization for semi-supervised progressive representation learning.
- **DealMvc** Yang et al. (2023): It proposes a novel dual-contrast calibration network for multi-view clustering, addressing the shortcoming of existing models that overlook similar samples across views.
- **ICMVC** Chao et al. (2024): It leverages multi-view consistency relation transfer and graph convolutional networks to handle missing values, and combines instance-level attention fusion with high-confidence guidance to jointly optimize multi-view representation learning and clustering performance.
- **DIVIDE** Lu et al. (2024): a decoupled, robust contrastive multi-view clustering method that identifies data pairs via high-order random walks to address false negatives and false positives.

- **PDMC-RCL** Lou et al. (2025): It quantifies the reliability of modality pairs using reliable contrastive learning and weights them accordingly, prioritizing the learning of discriminative features from reliable pairs while performing multi-level contrastive learning at both the feature and clustering levels.
- **CCMVC** Shi et al. (2025): It conducts joint training via contrastive learning at the feature, cluster, and view levels, and incorporates an alignment mechanism to ensure cross-view information consistency.

A.9   VISUAL ANALYSIS

To intuitively demonstrate the clustering performance of the GIB method on the dataset, we conducted a detailed visual analysis, with the results shown in Fig. 6, Fig. 7, and Fig. 8. Specifically, we used T-SNE to visualize the clustering results at different stages of the training process, namely the early, mid, and late stages, corresponding to the 0th, 5th, and 100th epochs of training. As the number of epochs increased, data points belonging to the same category gradually moved closer together, forming tighter and more cohesive clusters. Meanwhile, the boundaries between different categories became increasingly pronounced, leading to clear separations in the feature space. This phenomenon highlights the effectiveness of GIB in enhancing clustering performance.

