# OpenReview forum: "Granular Information Bottleneck for Deep Multi-modal Clustering"
_ICLR.cc/2026/Conference — ICLR 2026 Conference Withdrawn Submission_

### Official Review · Reviewer_N7yZ · 2025-10-29

**Soundness:** 2
**Presentation:** 2
**Contribution:** 2
**Rating:** 4
**Confidence:** 4

**Summary:**

This paper proposes a new framework for deep multimodal clustering called GIB (Granular Information Bottleneck). The core idea is to adaptively merge neighboring samples into “granular balls” instead of modeling each sample individually. Clustering is first performed on the centers of these granular balls, and the clustering results are then propagated back to the sample level, achieving a dual-layer clustering approach: “granular ball layer + sample layer.” The method introduces information bottleneck objectives at both levels: compressing redundant noise within each modality to retain only clustering-relevant information, while promoting consistency across modalities under the same clustering semantics. The authors reformulate these mutual information objectives into optimizable losses via variational approximations, combining them with Deep Divergence-based Clustering (DDC)-style clustering separation losses to derive the final training objective.

**Strengths:**

1. This paper employs “granules” rather than “individual samples” as the modeling unit, first aggregating similar samples into clusters before alignment. This approach smooths local noise while significantly reducing the alignment scale, directly addressing practical bottlenecks.
 2. The article features a well-designed structure. The methodology introduces two levels of information bottlenecks: the granule layer and the sample layer. The granular layer compresses redundant features across modalities in the feature space while ensuring global representation sharing. The sample layer enforces consistent category judgments across modalities during final clustering assignment, aligning with the fusion results. This simultaneously guarantees that “learned representations are compatible” and “final clusters are consistent,” aligning with unsupervised clustering objectives.

**Weaknesses:**

1. The stability of cluster structures remains to be verified. Clusters are progressively refined using pseudo-labels, a process dependent on the pseudo-label purity threshold, the maximum sample count threshold n_th, and the initial pseudo-labels obtained via weighted k-means. If the initial pseudo-labels themselves are unreliable, they may solidify erroneous structures, leading to over-segmentation or misclassification.
 2.A gap persists between theoretical foundations and practical outcomes. The paper approximates mutual information-related objectives using variational lower bounds and KL divergence, which rely on assumptions like Gaussian distribution and uniform prior distributions. These are empirical assumptions rather than rigorously guaranteed prerequisites, and the paper does not demonstrate whether performance would significantly degrade if these assumptions were relaxed.
3. The notation system is somewhat cumbersome, raising the reading threshold. The paper extensively uses subscripts and superscripts to distinguish between different entities like intra-modal representations, fusion representations, granule centers, and cluster assignments. However, these symbols sometimes carry multiple meanings in different contexts. For readers unfamiliar with this naming convention, this significantly increases the comprehension burden.

**Questions:**

1. Regarding the number of granules and complexity metrics. We hope the authors can report the final number of granules p obtained on the dataset, the ratio p/m of granules to original samples, and the actual differences in training time and memory consumption compared to the typical practice of sample-by-sample alignment. This would transform the point that p<<m from a qualitative statement into quantitative evidence, clearly demonstrating the method's real computational and memory benefits.
2. Regarding sensitivity to thresholds \theta and n_th. While the paper demonstrates the recursive segmentation process of granules, more systematic hyperparameter sensitivity results are desired. For example, how do ACC and NMI vary when the grain purity threshold \theta is adjusted between 0.7 and 0.95, or when the maximum grain capacity n_th is modified? How does the final number of generated grains change? Providing these curves or tables would directly demonstrate whether the method relies on finely tuned thresholds or remains stable over a broader range.
3. Regarding the impact of pseudo-label noise. Granule segmentation relies entirely on pseudo-labels. If the initial pseudo-labels themselves are erroneous, could this embed incorrect structures into subsequent training, leading to over-segmentation or mis-segmentation? We hope the authors clarify whether the current implementation recalculates pseudo-labels, resegments, or merges granules during the middle or late stages of training, or if the initial segmentation is fixed.

---

### Official Review · Reviewer_scxS · 2025-10-30

**Soundness:** 3
**Presentation:** 3
**Contribution:** 3
**Rating:** 4
**Confidence:** 4

**Summary:**

the paper presents a novel integration of granular computing and information bottleneck theory into the multi-modal clustering domain. The motivation is clear (Sec. 1), and the formulation is mathematically sound (Eqs. 1–10). Experiments (Table 1, Table 2) demonstrate consistent improvements in ACC/NMI metrics. However, several weaknesses limit its impact: (i) insufficient clarity in theoretical derivations and notation (Sec. 2.7–A.4); (ii) limited ablation and sensitivity analyses for key parameters (α, β, θ); and (iii) lack of insight into computational complexity and scalability. Despite these issues, the approach is original and promising. Overall evaluation: technically solid but requires refinement in analysis and clarity.

**Strengths:**

The paper introduces a novel and well-motivated framework that combines  granular computing and the information bottleneck principle for deep multi-modal clustering. This integration is original and effectively addresses the inefficiency and noise sensitivity of existing sample-level DMC models. The proposed dual-level bottleneck mechanism jointly optimizes granular and sample-level objectives, improving both feature discrimination and cross-modal alignment. Empirical results on five benchmark datasets demonstrate consistent and significant improvements over 19 strong baselines, confirming the method’s effectiveness and generality. The paper is also technically solid, with detailed proofs (A.2–A.4), convergence analyses (Fig. 5), and transparent experimental settings, making it both innovative and reproducible.

**Weaknesses:**

The transition from Eq. (8) to Theorem 2 lacks explicit intermediate steps and assumptions for the variational MI approximation; several symbols (e.g., ⊮, bn, M, q(x1)) are used before being consistently defined (Sec. 2.7; A.2–A.4; Eqs. (8)–(10)). This weakens rigor and reproducibility.

Parameter analysis only scans α, β (Fig. 4); purity θ and max-size nth are fixed (θ=0.9, nth=1.0) without sensitivity study (Implementation Details). Claims of robustness are not stress-tested under missing/noisy modalities; incompleteness is deferred to future work (Conclusion).

the extra cost from LDDC and dual-level MI terms is not analyzed (Sec. 2.6; Table 1 contains accuracy only).

Algorithm 1 stays high-level (e.g., “generate granular balls,” “jointly optimize”), without stopping criteria, gradient flow, or back-projection details; the fusion update S=∑wiHi (Eq. 27) lacks constraints/implementation guidance (A.5–A.6).

**Questions:**

Could you provide a step-by-step derivation from Eq. (8) to Theorem 2, clearly stating distributional assumptions and the role of the uniform constraint on q(x1)? Also, please define all symbols upon first use (⊮, bn, M, q).

How sensitive are results to θ and nth? Please add curves similar to Fig. 4 and discuss how θ,nth influence the number/quality of granular balls and final ACC/NMI.

What are training/inference time, peak memory, and FLOPs vs. leading baselines on IAPR/MIRFlickr? How do costs scale with #modalities m and #granules p (analytical complexity and empirical plots)?

Can you evaluate (a) missing-view rates (e.g., 30–70%), (b) noisy modalities, and (c) stability to pseudo-label initialization (multiple restarts or perturbations) for the splitting in Sec. 2.4.1?

Please detail (a) the granular-ball generation in latent space (initialization, split criterion, stopping), (b) sample↔granule mapping/back-projection used for LSamples, and (c) the update rule/constraints for wi in Eq. (27) (e.g., simplex projection, per-epoch schedule).

Since code is promised upon acceptance, is it possible to share a minimal reproduction package (or pseudocode with exact hyperparameters) during rebuttal to facilitate verification?

---

### Official Review · Reviewer_YiyL · 2025-10-30

**Soundness:** 3
**Presentation:** 3
**Contribution:** 4
**Rating:** 8
**Confidence:** 5

**Summary:**

This paper addresses the problems of existing deep multi-modal clustering methods, which rely on fine-grained input samples, exhibit weak robustness to noise, and also suffer from inefficiency. The authors propose a novel Granular Information Bottleneck framework, called GIB, which introduces granularity computation into multi-modal clustering and use granularity balls as the basic representation units. It designs a two-level information bottleneck constraint mechanism at the granularity level and the sample level, which jointly optimizes feature representations and clustering assignments through information compression and preservation. Experimental results show that GIB achieves performance superior to existing state-of-the-art methods on several datasets.

**Strengths:**

1.The paper introduces the concept of granularity balls as the input units for clustering. This granularity-based representation effectively filters noise and enhances model robustness, especially when dealing with high-dimensional and noisy data.

2.The paper explicitly draws on the Information Bottleneck principle and uses variational optimization to handle mutual information, providing strong theoretical backing. The derivations of theorems and propositions demonstrate the method’s rigor.

3.The method designs a novel dual-layer information bottleneck mechanism, performing information compression and preservation at the granularity level and at the sample level. This multi-level information constraint helps to jointly optimize feature representations and clustering assignments, enabling more fine-grained feature learning and better clustering alignment.

4.On multiple public datasets, GIB achieves significant improvements in ACC and NMI, outperforming several baselines, with particularly strong performance on the Caltech-2V, Caltech-3V, and WVU datasets. This demonstrates the method’s effectiveness in improving clustering accuracy and reliability.

**Weaknesses:**

1.Although the paper proposes a generation process for granularity balls (Fig. 3), the choice of the “optimal” granularity (i.e., the values of  n_{th} and θ) still relies on empirical selection and hyper-parameter tuning. Have the authors considered to make the generation of granularity balls more adaptive or automatic to obtain optimal granularity.

2.The paper does not consider the scenario of incomplete multi-modal data, while in practical applications, multi-modal data is often incomplete. How to extend GIB to incomplete multi-modal data is a meaningful issue.

3.The paper introduces a variational optimization approach to approximate mutual information. Actually, exact mutual information calculation and effective approximations remain challenging in high-dimensional spaces. Have the authors considered other estimation approaches. Or are there any more suitable ones.

**Questions:**

1.What distance metric is used to calculate the center and radius of granular balls? And why is this metric chosen?

2.The paper mentions fixed thresholds for granularity-sphere purity and maximum number of samples as 0.9 and 1.0, respectively. What is the justification or basis for choosing these two values?

3.In fact, exact mutual information calculation and effective approximations remain challenging in high-dimensional spaces. Have the authors considered other estimation approaches? Are there any more suitable ones?

---

### Official Review · Reviewer_2VFe · 2025-11-01

**Soundness:** 2
**Presentation:** 2
**Contribution:** 2
**Rating:** 2
**Confidence:** 4

**Summary:**

The paper introduces Granular Information Bottleneck (GIB). for deep multi‑modal clustering. Instead of operating on individual samples, GIB builds and updates granular balls—coarse units that cover groups of samples—and places dual information‑bottleneck (IB) constraints at two levels  (i) a granular level that reasons over ball centers and (ii) a sample level that aligns per‑modality and global cluster assignments. The authors frame the full objective in mutual‑information terms and optimize it with a variational treatment. They claim this is the first use of granular computing in deep multi‑modal clustering and report accuracy/NMI improvements on five benchmark datasets.

**Strengths:**

Using balls as units and constraining information flow at granular and sample levels  is a clear conceptual step beyond point‑wise pipelines;    The pseudo‑purity criterion and stopping rules (θ, (n_{\text{th}})).   are spelled out with visuals (Fig. 3), making the procedure reproducible.  Reasonable ablations and sensitivity checks: The module‑wise ablation and (α, β) sweeps are helpful; the full model’s gains over DDC alone are consistent.

**Weaknesses:**

The paper **chooses the best of 20 runs using ACC**, which depends on ground‑truth labels and should not guide model selection in an unsupervised setting. This can inflate reported performance and undermines the fairness of comparisons. A stronger protocol would report mean±std over fixed seeds or select by an *unsupervised* criterion.

Objective tension at the sample level  .\mathcal L_{\text{Samples}}) minimizes pairwise MI (I(Y^i,Y^j)) between per‑modality assignments while **maximizing** each (I(Y^i,Y^s)) to a shared consensus (Y^s) (Eq. 6). Minimizing (I(Y^i,Y^j)) reduces cross‑modal consistency, seemingly at odds with the paper’s stated goal of “clustering assignment alignment.” The paper would benefit from a justification (e.g., arguing this discourages *redundancy* while (Y^s) preserves *agreement*). As written, the design choice is under‑motivated.

The MI optimization relies on a simple KL‑based inequality (Eq. 8/Theorem 1) without specifying the parametric forms (q(\cdot)) or estimators used (e.g., MINE/InfoNCE bounds), nor analyzing estimator bias/variance. The theoretical section adds little beyond a generic bound; more detail is needed to assess stability and correctness.

Efficiency is claimed, not measured.The paper argues (p\ll m) reduces computation (Fig. 2) but provides **no runtime/memory** comparisons, nor reports average number of balls (p) per dataset. Given the method’s added machinery (ball splitting, MI estimation), actual wall‑clock/throughput data are needed to substantiate the efficiency claim.

Baselines skew older.  Table 1 includes many pre‑2022 methods; despite citing very recent multi‑modal IB work in the related‑work section, the experimental table omits such contemporaries. Stronger comparisons against recent deep multi‑modal clustering/IB approaches would make the “SOTA” claim more convincing.

6. *The method assumes complete multi‑modal observations; the conclusion explicitly notes this limitation and defers incomplete‑view handling to future work. This is important in practical deployments.

**Questions:**

Report means/standard deviations across fixed seeds and  remove ACC‑based model selection; if selection is required, use an unsupervised criterion.

Runtime & memory comparisons (and average (p)) against strong baselines to corroborate efficiency claims.

Ablate the MI terms in (\mathcal L_{\text{Samples}}): compare minimizing (I(Y^i,Y^j)) vs. maximizing it, or replace with a direct consensus‑regularization term; explain the rationale with theory or controlled experiments.

Compare to recent multi‑modal IB/contrastive methods cited in related work to substantiate SOTA claims.
*
Analyze θ and (n_{\text{th}}) sensitivity and visualize the evolution of ball counts (p) during training.

---

### Official Review · Reviewer_Aeuf · 2025-11-01

**Soundness:** 3
**Presentation:** 3
**Contribution:** 2
**Rating:** 4
**Confidence:** 5

**Summary:**

This paper proposes a novel Granular Information Bottleneck framework for deep multi-modal clustering. Its main idea is to replace sample level inputs with granular balls and develops a dual level information bottleneck mechanism that operates at both the sample level and the granular level. Experiments on five benchmark datasets demonstrate consistent improvements over 19 baselines.

**Strengths:**

1.	This paper is well-written and technically solid;
2.	The studied problem shows significance in multimodal clustering.

**Weaknesses:**

1.	This work appears to be similar to [1], including parts of the formulas, the optimization process. It is unclear why [1] was not cited or used as a baseline for comparison. After checking [1], it seems that the performance reported in this paper is not better than that of [1]. For example, on the IAPR dataset, the results are 50.6 vs. 52.9 and 25.4 vs. 28.7 in terms of ACC and NMI (this paper vs. [1]). It seems that the proposed changes do not bring improvements compared to [1] which limits the significance of this work.

2.	Although the paper claims efficiency gains, it lacks runtime or complexity comparisons versus sample-based DMC methods.

[1] Lou, Zhengzheng, et al. "Super Deep Contrastive Information Bottleneck for Multi-modal Clustering." Forty-second International Conference on Machine Learning.

**Questions:**

1.	What is the computational complexity of the granular splitting process compared to standard clustering?

2.	How sensitive is performance to the pseudo-label generation via weighted K-Means? Would random initialization degrade results?

3.	How should the purity threshold and the granular ball quantity threshold be set? How much do different values affect the model’s performance? Can the granularity level be learned in an end-to-end manner rather than being fixed by predefined thresholds?

---

### Note · Authors · 2025-12-23

I have read and agree with the venue's withdrawal policy on behalf of myself and my co-authors.